# Genotype- and Age-Dependent Differences in Ultrasound Vocalizations of SPRED2 Mutant Mice Revealed by Machine Deep Learning

**DOI:** 10.3390/brainsci11101365

**Published:** 2021-10-17

**Authors:** Denis Hepbasli, Sina Gredy, Melanie Ullrich, Amelie Reigl, Marco Abeßer, Thomas Raabe, Kai Schuh

**Affiliations:** 1Institute of Physiology I, University Wuerzburg, Roentgenring 9, 97070 Wuerzburg, Germany; sina.kollert@uni-wuerzburg.de (S.G.); amelie.reigl@uni-wuerzburg.de (A.R.); marco.abesser@uni-wuerzburg.de (M.A.); 2Center for Rare Diseases, University Clinic Wuerzburg, Josef-Schneider-Strasse 2, 97080 Wuerzburg, Germany; ullrich_m@ukw.de; 3Center for Medical Informatics, University Clinic Wuerzburg, Schweinfurter Strasse 4, 97080 Wuerzburg, Germany; 4Institute for Medical Radiation and Cell Research, Campus Hubland, University Wuerzburg, Biozentrum, 97074 Wuerzburg, Germany; thomas.raabe@uni-wuerzburg.de

**Keywords:** SPRED, SPRED2, mice, neural networks, ultrasound vocalizations, DeepSqueak

## Abstract

Vocalization is an important part of social communication, not only for humans but also for mice. Here, we show in a mouse model that functional deficiency of Sprouty-related EVH1 domain-containing 2 (SPRED2), a protein ubiquitously expressed in the brain, causes differences in social ultrasound vocalizations (USVs), using an uncomplicated and reliable experimental setting of a short meeting of two individuals. SPRED2 mutant mice show an OCD-like behaviour, accompanied by an increased release of stress hormones from the hypothalamic–pituitary–adrenal axis, both factors probably influencing USV usage. To determine genotype-related differences in USV usage, we analyzed call rate, subtype profile, and acoustic parameters (i.e., duration, bandwidth, and mean peak frequency) in young and old SPRED2-KO mice. We recorded USVs of interacting male and female mice, and analyzed the calls with the deep-learning DeepSqueak software, which was trained to recognize and categorize the emitted USVs. Our findings provide the first classification of SPRED2-KO vs. wild-type mouse USVs using neural networks and reveal significant differences in their development and use of calls. Our results show, first, that simple experimental settings in combination with deep learning are successful at identifying genotype-dependent USV usage and, second, that SPRED2 deficiency negatively affects the vocalization usage and social communication of mice.

## 1. Introduction

Animal vocal communication is a widespread phenomenon that provides insight into context-dependent signaling and reception [1,2,3]. Similarly, mice engage in social communications through a rich repertoire of ultrasonic vocalizations (USV) and their calls vary in frequency and length. Most are in the range of 30–115 kHz and are about 2–50 ms long [4,5]. Although the calls seem simple, at second glance they are structured and recurring. So far, it has been assumed that females rarely vocalize while being together with a male [6,7]. Newer studies suggest that females vocalize in same-sex interactions [8,9], with males during mating [10], and prefer males which vocalize, as well [11]. In contrast, male mice show increased vocalization during the presence of a female or while mating [12]. It has also been shown that males change their syntax depending on whether a female is in the vicinity or not [13]. Mice calls have already been divided into categories and attempts have been made to analyze them further, but no uniform system has been established, till today [14,15]. First efforts were made to assess the meaning of USVs in rats and mice as well as to interpret these calls as an indicator of the subjective experience of the animal. For instance, in rats it was shown that a high frequency, around 50 kHz, is associated with positive events, and a lower frequency, around 22 kHz, with negative events [16,17]. Interpretation of mouse calls is not that far advanced, but their call usage has already been examined in various social situations. USVs of young and old mice were examined during same-sex interactions but also between male–female interactions [12,18]. This type of male–female interaction test is widely used to study vocal phenotypes in male mice [6,19,20].

In previous studies, SPRED2-knockout (KO) mice showed clear signs of neurohumoral disturbances [21] and OCD-like (obsessive compulsive disorder) behavior [22]. Since language problems (dysphonia, dysarthria) are related to developmental diseases, neurodegeneration, and advancing age [23], we wanted to examine the call usage to demonstrate developmental delay and neurodegeneration through a different approach. A more detailed understanding of the USV spectrum may become an additional valuable parameter, for example, in analyses of phenotypes of genetically modified mouse models. Until now, this was associated with a huge effort due to the collection of a large number of very short calls in long time periods. Analyzing this large amount of data manually required a lot of work force. This might provide one reason why not many data were collected, which could be used to decode the vocal communication of mice. 

To identify differences in USV usage depending on the genetic manipulation of the SPRED2 gene in mice in comparison to WT control mice, we used the DeepSqueak program [24] to search for and classify all calls by using a deep-learning convolutional neural network (CNN) specifically adapted to this project.

## 2. Materials and Methods

### 2.1. SPRED2 Mutant Mice

SPRED2-KO mice were generated by a gene trap approach as described previously [25]. In brief, the SPRED2 gene was disrupted by insertion of a gene trap vector, resulting in a disrupted, non-functional *Spred2* gene and the expression of a truncated, dysfunctional SPRED2 protein [22]. For our recordings, we used either mice with a homozygous KO in both alleles or bi-allelic SPRED2 wild-type (WT) mice as controls. All mice used in this work were raised and kept in our in-house mouse facility under standard conditions according to the guidelines of the European Union (2010/63/EU). Mice were housed in a 12/12 h light–dark cycle under controlled room temperature (21 ± 1 °C) and humidity (55 ± 5%). They had access to tap water and standard mouse chow ad libitum. To minimize possible inbred effects, mice were bred in a mixed 129/Ola x C57Bl/6 genetic background and not crossed back to a specific inbreed background. In all experiments, we used pairs of SPRED2 mutant mice and pairs of WT mice as controls, which were not littermates or siblings and were naïve to male/female social encounters. Mutant and WT littermate mice were identified as described before [22]. The young groups of mice were between 3 and 6 months old, the old groups of animals were one year and older. Mice were divided into gender groups after weaning and thus had no further contact with the opposite sex before the experiment. 

### 2.2. Recording and Hardware 

Testing took place in a clean, empty cage (Eurostandard Type II, 268 × 215 × 141 mm, Tecniplast, Italy) without embedding. To reduce background noise, the cage was placed into a self-built 50 × 50 × 50 cm sound-isolating box (Foam: aixFoam SH001 MH). The ultrasound microphone (Pettersson Elektronik AB, Sweden, M500-384) with a frequency response range of 10 to 210 kHz was securely placed above the cage with a distance between 20 and 25 cm from the mice during recording. They were either recorded with the “BatSoundTouchLite” or “Audacity” program on a Lenovo ThinkPad T570, or with the App “Bat Recorder” and a Huawei T1-A21L Tablet. All recordings took place in the same time period during the light cycle since we wanted to investigate genotypic differences in the communication of mice in a social context of mice and not during mating.

### 2.3. Experimental Setup “Speed Dating” 

Before using the females in the “Speed Dating” experiments, their estrous cycle was evaluated visually based on criteria described by Champlin et al. [26], as well as with the vaginal cytology method [27]. We attempted to use mice in the proestrus stage of the estrous cycle, to achieve a comparable starting point in all groups. Since we only wanted to motivate the males to emit calls and not analyze specific mating calls, this was a sufficient experimental setup for our study. Females who were not in this stage of the estrous cycle at the time of recording were first excluded from the experiment and included later.

All recordings took place in the sound-isolating box described above. In total, one recording cycle lasted 15 min. In the first five minutes, only the female mouse was placed into the empty cage to settle in. After these five minutes, the male mouse was added and the recording was started. Calls that were emitted during this time were included in the analysis. After the five-minute recording period, the male was removed. In the last five minutes of recording, only the female was left in the cage to prove the muteness of females. No calls were detected in this last period and no calls could be included in the analyses. 

Only WT/WT and KO/KO couples were put together. Each cage was replaced before the next recording. A total of 16 young WT mice (8 male/8 female, from 13 different litters), 9 old WT mice (3 male/6 female, 6 litters), 14 young SPRED2-KO mice (8 male/6 female, 12 litters) and 7 old SPRED2-KO mice (3 male/4 female, 6 litters) were used. The mean age of the mice within each group was 107 days for the WT young mice, 127 days for the KO young mice, 506 days for the WT old mice and 478 days for the KO old mice. Each mouse had at least 30 min of rest alone in its original cage before the next recording cycle.

### 2.4. Call Analysis and Classification

Acoustical analysis of the recordings was performed by a specific neural network using DeepSqueak (Version 2.5.0 and 2.6.2) [24]. To improve the detection rate of DeepSqueak, we used over 8000 spectrogram images of our own recordings to program a custom-trained Faster-RCNN object detector neural network, which was trained specifically to the USVs of our mice. Based on our custom-trained CNN and verified by scrolling through the spectrograms (DeepSqueak 3.0), we lowered the rate of missed calls to 8.54%, which was much lower than the previously reported rate of missed calls (27.13%) of the standard DeepSqueak CNN [28].

Combining our DeepSqueak results and previously published USVs [14,15,29], we grouped and classified all calls emitted by our mice in 10 different classes (Table 1). Classification of calls was performed unsupervised by a second Faster-RCNN neural network, which was trained to the USVs of mice in our in-house facility. False-positive sounds generated by the cage or misclassified calls were excluded manually. Less than 1% of the recorded sounds could not be classified because they did not fit in any of the categories above and were typically sounds generated by the movements of mice. Transition probabilities were calculated by the DeepSqueak function Syntax Analysis [24]. Each call was examined further in terms of four characteristics: call length (in sec.), principal frequency (in kHz), delta frequency (in kHz) and mean power (in dB/Hz). We used DeepSqueak to arrange the calls as T-SNE (T-distributed Stochastic Neighbour Embedding) plots to visualize the complex and multi-factor dependent data in two dimensions, sorted by frequency.

### 2.5. Statistics

Only USVs that occurred during the 5 min when males and females met were selected. Calls that occurred less than 1% or could not be classified were excluded. Data were analyzed in various ways. Two-way mixed ANOVA (Figure 1k) with genotype (WT/KO) and age as between-subject factors was carried out in IBM SPSS 21. Significance tests between two groups were carried out in two-tailed two-sample t-tests in GraphPad Prism 5.0. A *p*-value of 0.05 or smaller was considered statistically significant (Figure 2e and Figure 3e–h).

Two-way mixed ANOVA in Figure 4 was done in IBM SPSS 21. Each call type was checked separately with Bonferroni correction in four cases, resulting in a significance level of *p* = 0.0125 or smaller. Significant differences are marked by an asterisk. The analyses and figures of the transition probability (Figure 3a–d) and the T-SNE plots (Figure 5) were calculated and created directly in DeepSqueak.

## 3. Results

### 3.1. Genotype- and Age-Dependent Call Classification

To record and analyze the calls from WT and KO mice, we chose a standard situation described previously [12,19], which was designated as “Speed Dating”. After initial analyses and adjustments to the algorithm, we divided the calls into ten major groups (examples in Figure 1a–j), which were similar to groups of calls described before [14,15]. With these calls, we then trained the deep-learning algorithm in DeepSqueak to specifically recognize and classify the calls of mice from our mouse facility.

During the five-minute speed date recordings, the averaged sum of all emitted calls of young (3–6 months) and aged KO (one year and older) mice was lower than the sum of USVs emitted by age-matched WT mice control groups. Furthermore, the number of total USVs increased age dependently (Figure 1k, genotype: *p* ≤ 0.05, age: *p* = 0.045).

Analysis and classification of all recorded calls is summarized for all four groups (WT Young, WT Old, KO Young, KO Old) in pie charts (Figure 2a–d). The pie charts include all call classes that made up more than 1% of the total number of calls. Comparing the call usage of WT Young vs. KO Young, we noticed that it was very similar for some call classes, e.g., Inverted U, Frequency Steps, and Down. However, Step Up calls were hardly detectable in WT mice groups (below 1%), but almost exclusively detected in KO mice pairings (Figure 2c–e). The Step Up call occurred also more frequently in the old than in the young KO group (Figure 2c,d).

### 3.2. Genotype-Dependent Call Syntax Differences

In order to detect which call combinations were used more frequently than others, we analyzed the call order probability, i.e., if a specific call follows a certain predecessor. Thereby, it was possible to investigate not only the distribution of calls but also to examine which call combinations appeared more often than others, as well as to closely monitor the complexity and syntax of the mouse vocal repertoire.

All calls that occurred in the respective test groups with a usage probability of more than 1% are displayed in Figure 3. The values in the individual boxes indicate the probability of one call following the previous. The following call is indicated on the *x*-axis. At first glance, some call sequences seemed similar in all groups. For example, the call Frequency Step was followed by another Frequency Step call with a high probability in all four groups. In contrast, the use of the Short call as a follow-up call, which was rarely used by the WT (Figure 3a,b), was more often used by KO mice (Figure 3c,d). Older mice seemed to display a more homogeneous distribution of possible combinations. The Flat calls were used significantly more often as follow-up calls by young KO as compared to young WT mice, but there were no significant differences in usage of the Flat call as follow-up in the old test groups (Figure 3e–h; WT young vs. KO young: *p* = 0.0061). This indicated that the young mice, which were already sexually mature at the age of 3–6 months, continued to develop their vocal repertoire and learned new combinations over the course of their lives.

### 3.3. Genotype- and Age-Dependent Differences in Call Characteristics

Significant genotype- and age-dependent differences in USV usage were identified by two-way ANOVAs. Figure 4a–d shows significant differences in various parameters, which were traced back to genotypes. The average call length across all USV subtypes of KO mice was significantly shorter as compared to WT calls, in young as well as in older animals (Figure 4a; *p* < 0.001). When using the Flat and Step Down calls, the average frequency at which the mice communicate with each other was significantly higher in the WT group compared to KO mice (Figure 4b; *p* < 0.001, Figure 4c; *p* < 0.001).

However, the mice also showed age-dependent differences, such as the mean power of the Step Down call (Figure 4e; *p* = 0.001) or the call length of Down (Figure 4g; *p* < 0.001). Some of the age-dependent differences in the call variations between young WT and KO animals became smaller with age. Obviously, the SPRED2-KO mice showed a development in which their call usage came closer to the WT. Although we did not observe one parameter in which the old KO animals reached the same values as the old WT animals, a trend in this development was noticed. This trend was seen, for example, in the categories of average volume (mean power) and call length of the calls Down and Step Down, in which the KO mice approached the WT mice over the course of their age (Figure 4e–g).

### 3.4. T-SNE Plot Visualization

In order to provide an overview of our collected data, we used DeepSqueak to arrange the calls as T-SNE plots sorted by frequency.

Figure 5a displays all calls (*n* = 1658) of young WT mice in the 5 min meetings of males with females. All Step Down calls were grouped in the upper left corner. The UPs formed a clear cluster in the upper middle area and the Shorts were all collected in the lower area of the picture. Obviously, there were clear clusters of the individual calls in the young WT animals and thus clearly assigned frequency ranges. In Figure 5c, old WT (*n* = 3326) show a different picture as compared to young WT mice. Although more calls were detected here, there were no clearly separated clusters anymore. Individual calls were grouped in certain regions, as well, but the transitions between the groups seemed more contourless. This, again, showed the changes in USV usage of mice over the course of their life. Figure 5b shows the T-SNE plot of the young KO calls (*n* = 238). Although no clear statement could be made due to the low number of detected calls, some smaller groups of call classes were seen. However, Figure 5d (*n* = 1271) shows again clear similarities between old KO and old WT calls. Here, individual call clusters were also arranged but with diffuser transitions and more homogeneously distributed calls. This phenomenon of more homogeneity is obvious in general, especially in T-SNE plots of WT mice, independent of age. Taken together, T-SNE plots again reflected age- and genotype-dependent differences in usage of USVs, call frequencies and the changes in USV usage in mice.

## 4. Discussion

Language problems are often related to developmental or neurodegenerative diseases. Based on previous findings, which showed a neuronal dysfunction in SPRED2-KO mice, we wanted to examine their call usage to investigate genotype-dependent differences in this mouse model. Using DeepSqueak [24], we evaluated and analyzed over 150 h of recording material for this study and found certain call profiles and significant genotype-dependent differences between WT and SPRED2 mutant mice. For these investigations, we chose the situation of the first meeting between males and females, a common and straightforward experimental setup, which was suggested previously [12,15].

We detected clear differences in the total number and the composition of calls between WT and KO mice. For example, the Step Up call was used almost exclusively by KO animals. Although the meaning of the individual calls is still unclear, we observed interesting and newfound results. All calls of the mice emitted in this given situation could in most cases be divided into the 10 characterized call categories (Figure 1a–j), which is in line with earlier studies, where calls were classified into 9–14 different syllable types [4,12,14,29]. This indicates that mice do not just randomly emit sounds and that their sounds have a meaning and follow rules.

A significant difference occurred when we compared the total number of calls (Figure 2). Our results show that SPRED2-KO mutant mice were influenced in their communication development, as shown by changes in USV usage during aging. Studies have shown that female mice spend more time with vocalizing males than with mute ones [11]. Therefore, KO males could have a distinct disadvantage with females and mating, as they communicate significantly less compared to the WT mice. In this regard, further investigations should be carried out to identify possible deficits in mutant mice. 

Striking was the significantly higher number of Step Up calls in the KO mice (Figure 2e). Unfortunately, the meaning of this call could not be retrieved. However, a clear difference in the vocal repertoire of SPRED2-KO mice was seen here again. A further assignment of calls to certain events should give us a more detailed insight into the context of these differences in USV usage. 

We used call syntax analysis to further investigate the development of communication complexity. There are many studies on the vocalizations and syntax of other mammals and birds, but not much is known about the syntax in the vocalization of mice. Holy and Guo [4] introduced the idea that mice vocalizations have some features similar to the courtship songs of songbirds. Newer studies showed that male mice do change their repertoire composition and use different syntax, dependent on the social context or presence of female mice [12,13,30]. By analysis of the probability of a certain call following the preceding call, we investigated which call combinations were used more often. As depicted in Figure 3, a trend that communication develops throughout the animals’ lives seems obvious. Older mice used some combinations not used by young animals. This is rather more evident in the Long calls such as Frequency Step or Complex than in the Short calls (Short, Flat and Up), and is particularly noticeable with Frequency Step calls following Frequency Step or Complex. 

All these changes in USV usage clearly show the development of communication throughout the life of mice. Older animals not only emit a higher number of calls, but they might have a more complex vocalization, as well. We cannot say with certainty that longer calls automatically stand for a more complex communication. It may also stand for a deterioration of the vocal system of mice. However, it is clear that there is a change in the composition of call usage.

This is an interesting finding since the mice are already fully grown at the age of 3–6 months and are sexually mature at the age of about six weeks. On the other hand, a previous study showed that males who mate with females or who are within their vicinity emit shorter and simpler calls than those that only smell female urine and thus produce longer and more complex calls [13]. 

Although we cannot determine the exact meaning of certain calls by the results of this study, we assume a development of communication with more complex combinations throughout the animals’ lives. There are very clear genotypic differences in the total number and percentage distribution of the calls. The call syntax analysis shows age-specific differences that indicate clear structures and processes in the communication of mice. 

In addition to the calls themselves, we examined the length, frequency and strength of individual calls. This revealed a number of significant changes between the four test groups. A reliable interpretation of how these individual differences affect communication in mice is hardly possible. We detected differences in all areas and categories, but some calls showed a higher variability in characteristics than others, for example, the call Down (Figure 4f,g), which was particularly variable in the four test groups. So far, very few data are available on this topic in the form of publications. Therefore, it is very difficult to assess what the individual calls or differences in length or frequency of a call mean. In sum, the data shown in Figure 3, Figure 4 and Figure 5 demonstrate a call development in the course of a mouse’s life, especially visible in the syntax analysis in Figure 3, which shows that communication develops independently of the genotypes investigated in this study. 

Our study revealed differences in vocal communication, some of which turned out to be age or genotype dependent, or both. In all categories, SPRED2-KO mice seemed to lag behind the WT mice both in their number of emitted calls and in the subtlety of how the individual calls are carried out. Surprisingly, we observed an interesting trend in the KO animals. While the differences between WT and KO were very clear, especially in the young animals, the KO mice seemed to adjust more and more to the WTs over the course of their life. One reason for this might be that KO animals have spent their entire lives in cages with WT and heterozygous mice and learned from them [31,32].

However, SPRED2-KO mice did not completely reach the level of WT mice, which could be explained by their phenotype, indicating physical and mental impairments such as OCD [22] and hormonal imbalance [21]. Furthermore, dysfunction of SPRED1, a homologous member of the SPRED protein family, causes cognitive deficits both in humans [33] and mice [34], altogether possible reasons for the impaired communication in SPRED2-KOs.

T-SNE plots enable the breaking down of very complex and multi-factor-dependent data, and its visualization in two dimensions. Although these plots are not suitable to visualize small details, they provide a good overview. The clearly defined clusters in WT mice T-SNE plots compared to KO mice T-SNE plots reflect a repeated usage and equal modulation of the calls in WT mice. In contrast, the less distinct grouping of KO mouse calls reflects that their calls were less delimitable from their other calls. The plots also show clearly, apart from the statistically relevant ANOVAs, how strongly WT and KO mice differ in the total number of calls. The very low number of total calls of the young KO group strengthens the hypothesis that they are not only physically underdeveloped in their first months, but also that their vocalization development is impaired.

Another observation was that the old KO group did not use all frequencies or the same frequency ranges as the WT mice. This again shows the significant vocalization impairment of the KO animals and that they are not able to reach the call profile of the WT animals. Therefore, the T-SNE plots visualize the differences in the call spectrum of WT and KO animals, and they reflect the development of the communication of mice over their lifetime.

## 5. Conclusions

Taken together, we were able to train the DeepSqueak neural networks to our mice specifically, and with that were able to successfully classify and analyze them automatically. Our results show that mice calls develop and vary over time and that SPRED2 deficiency negatively affects their social communication. Furthermore, our findings provide the first USVs classification of SPRED2-KO vs. WT mice and, to our knowledge, provides the first syntax analysis of mice over a longer period.

## Figures and Tables

**Figure 1 brainsci-11-01365-f001:**
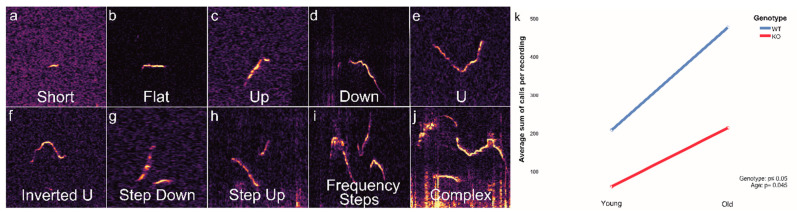
Subdivision of the ten different call types and their percentage distribution. (**a**–**j**) Examples of calls, generated by DeepSqueak. (**a**) Short. (**b**) Flat. (**c**) Up. (**d**) Down. (**e**) U. (**f**) Inverted U. (**g**) Step Down. (**h**) Step Up. (**i**) Frequency Steps. (**j**) Complex. (**k**) Total number of calls: two-way ANOVA demonstrating the averaged sum of calls per five-minute recording of young and old animals from WT and KO (WT Young *n* = 1658 ± 214, WT Old *n* = 3326 ± 358, KO Young *n* = 238 ± 40, KO Old *n* = 1271 ± 142; genotype *p* ≤ 0.05; age *p* = 0.045).

**Figure 2 brainsci-11-01365-f002:**
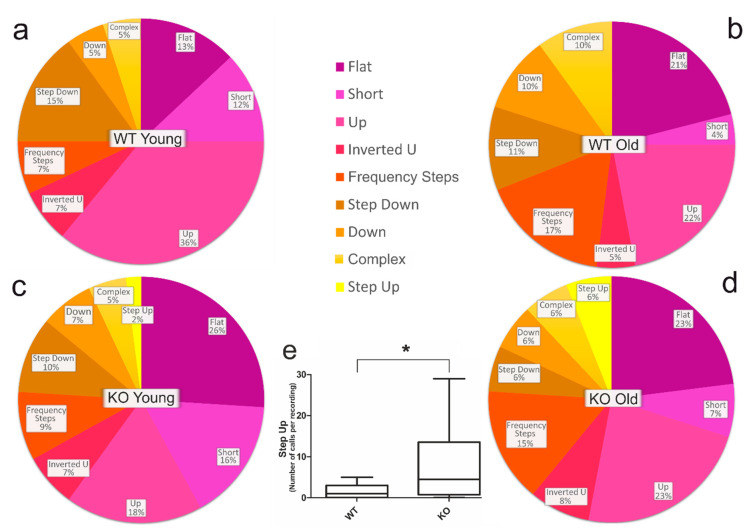
Relative number of calls. (**a**) Distribution of calls of young WT in percent (*n* = 1658). (**b**) Distribution of calls of old WT in percent (*n* = 3326). (**c**) Distribution of calls of young KO in percent (*n* = 238). (**d**) Distribution of calls of old KO in percent. (*n* = 1271). Calls with less than 1% (U) are excluded in all pie charts. (**e**) EUR number of Step Up calls: comparison of the total number of Step Up calls per recording (*p* = 0.0146). * *p*< 0.05.

**Figure 3 brainsci-11-01365-f003:**
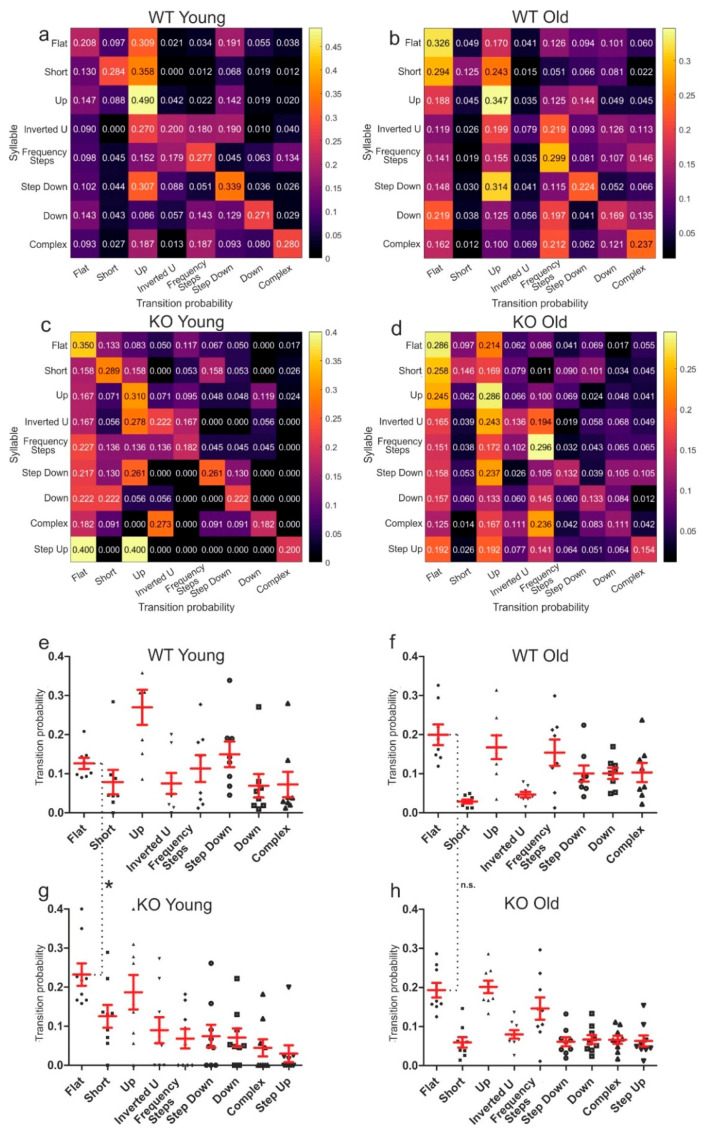
Syntax analysis of transition probabilities, indicating the relative transition prevalence of a certain call following a previous one. (**a**–**d**) The following call is indicated on the *x*-axis. (**a**) Transition probabilities of young WT mice. (**b**) Transition probabilities of old WT mice. (**c**) Transition probabilities of young KO mice. (**d**) Transition probabilities of old KO mice. (**e**) Dot plot of the transition probabilities of each WT young call. (**f**) Dot plot of the transition probabilities of calls emitted by the old WT mice. (**g**) Dot plot of the transition probabilities of each KO young call (* *p* = 0.0061). (**h**) Dot plot of the transition probabilities of calls emitted by the old KO mice.

**Figure 4 brainsci-11-01365-f004:**
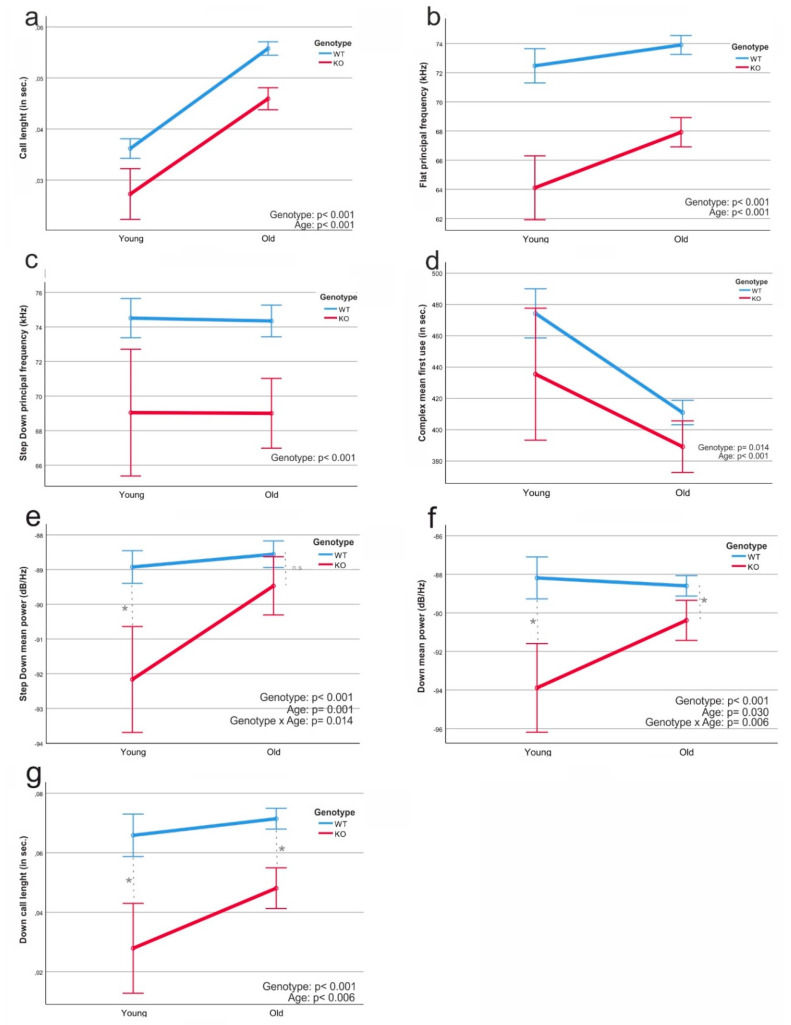
Two-way ANOVAs from different calls and factors of the four test groups. (**a**) Call length: shows the general length of all calls in seconds from WT and KO. (**b**) Flat, principal frequency: shows the development of the basic frequency of the Flat call from WT and KO. (**c**) Step Down, principal frequency: shows the development of the basic frequency of the Step Down call from WT and KO. (**d**) Complex, mean first use: shows the average use of the Complex call at the beginning of the encounter between males and females. (**e**–**g**) The development of vocalizations of the KO animals in the course of their life. (**e**) Step Down mean power: shows the development of the average strength (dB/Hz) of the call Step Down of the animals at a young and advanced age. (**f**) Down mean power: shows the development of the average strength of the call Down of the animals at a young and old age. (**g**) Down call length: Shows the development of the length of the call Down of the animals at a young and old age. *p* values as indicated in the figure. * *p*< 0.0125.

**Figure 5 brainsci-11-01365-f005:**
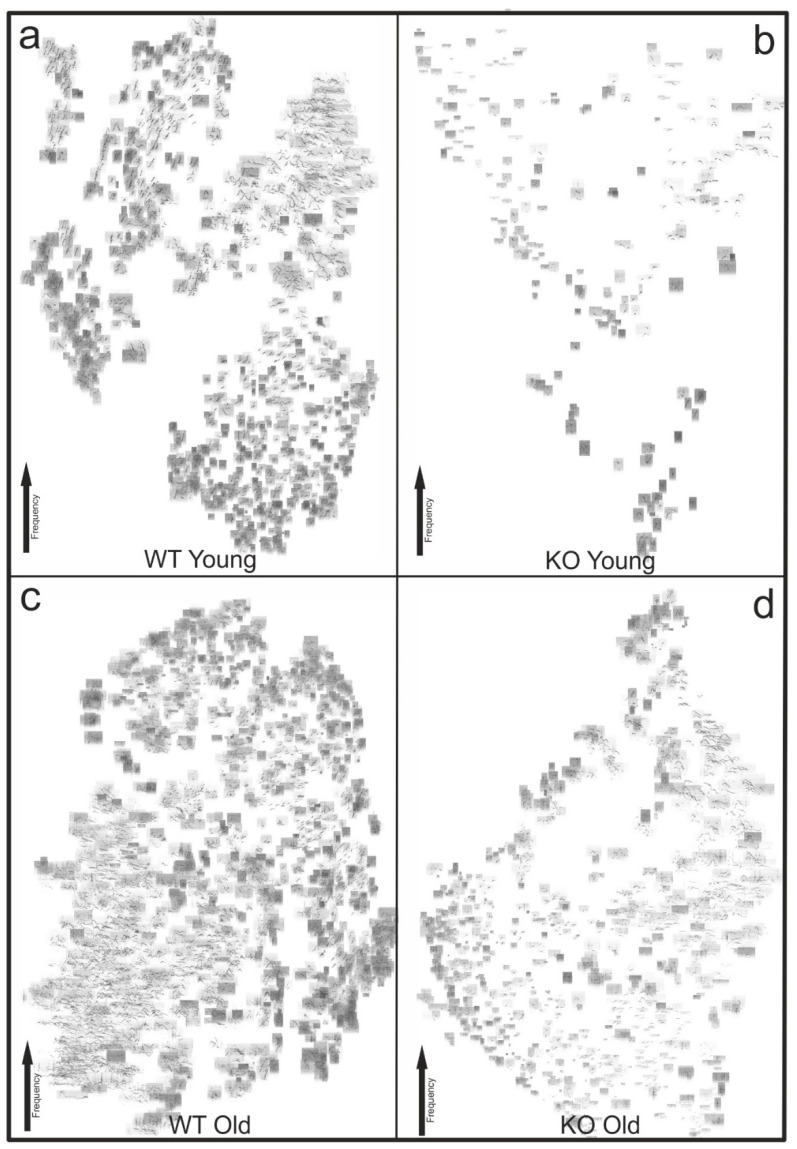
T-SNE plots of all four test groups. (**a**) T-SNE plot of all calls from young WT mice, plotted by frequency (*n* = 1658). (**b**) T-SNE plot of all calls from young KO mice, plotted by frequency (*n* = 238). (**c**) T-SNE plot of all calls from old WT mice, plotted by frequency (*n* = 3326). (**d**) T-SNE plot of all calls from old KO mice, plotted by frequency (*n* = 1271).

**Table 1 brainsci-11-01365-t001:** List of all 10 different call classes and their main point of difference.

Short	duration of less than 10 ms
Flat	longer than 10 ms but no large change in delta frequency
Up	monotonically increasing in frequency
Down	monotonically decreasing in frequency
U	U-shaped calls, frequency drops in the beginning but then rises again to about the same as it started
Inverted U	inverted version of the U call
Step Down	two calls which closely follow each other and have a frequency change to a lower frequency
Step Up	two calls which closely follow each other and have frequency a change to a higher frequency
Frequency Steps	long calls (>50 ms) with instantaneous frequency changes appearing as stepsbut with no interruption in time
Complex	long calls (>50 ms) with a straight course and variation in frequencyor calls with two different parallel frequencies at the same time (harmonic component)

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
