# Peer review of "Genotype- and Age-Dependent Differences in Ultrasound Vocalizations of SPRED2 Mutant Mice Revealed by Machine Deep Learning"

_brainsci, 2021, doi:10.3390/brainsci11101365_

Round 1
Reviewer 1 Report
This study evaluated how the ultrasonic vocalizations of Sprouty-related EVH1 do-14 main-containing 2 (SPRED2) deficient mice differed from wild-type. They found differences between groups in USV production rates and call classifications.
Strengths
The major strength of this study was that they used Deepsqueak to semi-automatically analyze and classify call categories between two genotypes and two ages. However, many weaknesses are present in this manuscript that require major revisions. Also, the discussion is overreaching and requires a new lens of interpretation.
Weaknesses
The overall paragraph, sentence structure, and spelling/grammar requires improvement prior to publication. Some major issues include the lack of transitional sentences (e.g., line 49-50), fragments (e.g., line 49-50), spelling mistakes (e.g, line 33), using abbreviations before defining (e.g., line 74), and the disorganization of the introduction which lacks a central hypothesis.
Regarding the methods, there is insufficient details regarding the female mouse estrous cycle, whether the mice were recorded during the dark portion of the light cycle, and the statistical methods planned.
Figures 1 and 4 show averages but there are no standard deviation bars.
Figure 2 the pie charts have 9 categories but the statistics below (e-g) have combined call categories with no rationale why there is a difference.
It is unclear if differences in acoustic parameters reported in line 189—193 are across all USVs or subtypes.
The term “language” throughout the document is inappropriate because we cannot compare human language to rat/mouse USVs based on this study. The repertoire of USV call subtypes does not equate to language or indicate rule following. Many animal species have a repertoire of vocalizations they produced but that itself does not constitute language. This comparison (especially in the discussion) is both overly simplifying language and overextending their research findings. In fact, figure 3 clearly demonstrates that there is little syntax existing in the emitted USVs. Also, it is assumed by the authors that new combinations of call types are a development of linguistic abilities in the older rodents; however, these new combinations could also be a deterioration of the system (i.e., forgetting the ‘rules’). The authors should revisit the linguistic literature prior to making such bold claims about animal ‘language’.
Author Response
First of all, we would like to thank the reviewers for the constructive and technically feasible points of criticism! We have addressed all queries raised by the reviewers and included the improvements in the revised manuscript. Please find here the detailed reply to your queries:
Reviewer #1:
Weaknesses
The overall paragraph, sentence structure, and spelling/grammar requires improvement prior to publication. Some major issues include the lack of transitional sentences (e.g., line 49-50), fragments (e.g., line 49-50), spelling mistakes (e.g, line 33), using abbreviations before defining (e.g., line 74), and the disorganization of the introduction which lacks a central hypothesis.
Reply: We have revised the entire manuscript and tried to eliminate all spelling and grammar mistakes. Before resubmission, the revised manuscript underwent a spell-check. A central hypothesis was added to the introduction.
Query 1: Regarding the methods, there is insufficient details regarding the female mouse estrous cycle, whether the mice were recorded during the dark portion of the light cycle, and the statistical methods planned. Figures 1 and 4 show averages but there are no standard deviation bars.
Reply: The requested details were added to the M&M section (lines 66 – 112). In Figure 1, the standard deviations were added to the legend (lines 160, 161). In Figure 4, the SD was included in the Figures (page 8).
Query 2: Figure 2 the pie charts have 9 categories but the statistics below (e-g) have combined call categories with no rationale why there is a difference.
Reply: We have deleted these misleading plots of combined categories and – because of their similarity - we included the original category Harmonic into the category Complex.
Query 3: It is unclear if differences in acoustic parameters reported in line 189—193 are across all USVs or subtypes.
Reply: We have included a more detailed description of these parameters, which differentiates better between all USVs and subtypes (lines 206 – 214).
Query 4: The term “language” throughout the document is inappropriate because we cannot compare human language to rat/mouse USVs based on this study. The repertoire of USV call subtypes does not equate to language or indicate rule following. Many animal species have a repertoire of vocalizations they produced but that itself does not constitute language. This comparison (especially in the discussion) is both overly simplifying language and overextending their research findings. In fact, figure 3 clearly demonstrates that there is little syntax existing in the emitted USVs. Also, it is assumed by the authors that new combinations of call types are a development of linguistic abilities in the older rodents; however, these new combinations could also be a deterioration of the system (i.e., forgetting the ‘rules’). The authors should revisit the linguistic literature prior to making such bold claims about animal ‘language’.
Reply: The term “language” was replaced throughout the manuscript and the far-reaching statements about language development in rodents were toned down.
Reviewer 2 Report
In the manuscript Hepbasli et al. examined the vocal repertoire of SPRED2 mutant mice in a male/female social interaction context. In previous studies the authors presented data suggesting changes in brain functions of the KO and related them to OCD-like behaviors. The authors used an automated machine-learning algorithm to discern specific types of vocalizations. While attempts of classifying USVs have been done for several years only few and very recent papers done this using computer learning and on some kind of mutant mice. Overall, the paper is comprehensive. Nevertheless, it requires major revisions and clarifying in some areas in the methodology and results presentation and minor adjustments to transparency and readability.
Major points:
- The description of the subjects used requires revision. A more thorough characterization of the genetic background of the animals would be beneficial. From where were the original breeding pairs mice obtained? Where were the mutants generated? How many crossings with C57Bl/6 mice were there before the experiment? Please specify whether the WT and KO mice used in the experiment were siblings as it is unclear from description. Please place the information about the age of the animals during the recording sessions in M&M. Describe if the mice were naïve or experienced in male/female social encounters as this significantly alters their vocal repertoire. This information is presented later in the paper however M&M should contain such details. Furthermore, were the females checked for estrous cycle since the state that they were in could affect the outcome of the experiment?
- Were the animals handled prior to the experiment? Were they familiarized with the “speed dating” setup? Did the analyzed recordings only include the 5 minutes when the male and female were together? In the described procedure the USV recordings should contain both male and female vocalizations at once, did the authors take that into consideration that the observed differences in groups may sex sensitive?
- The authors have used DeepSqueak to analyze the USVs, specifically versions 2.5.2 and 2.6, these versions of the program do not allow free scrolling of the spectrogram. Only the newest 3.0 version incorporated this option from a GitHub fork branch. In a recent paper by Fonseca et al. (DOI: 10.7554/eLife.59161) compared several machine learning software in detection of mouse USVs. In their study DeepSqueak had an astounding 27.13% rate of missed calls, which from personal experience I find plausible. The recordings should be checked for missed calls. The call classification was done unsupervised by a second neural network. What was the input data used to train the network, were those the same 8000 images used for detection? I also suggest that after automatic classification the USVs types should be evaluated by a human since software does not guarantee perfect assignment of USV types.
- In results multiple different variables are presented as proof of difference of KO mice. Please specify, how many parameter comparisons were made in total for each of the presented data? Why some calls in the presented results pulled together (flat, short, up) and others like step up are not, also complex and harmonic calls are once grouped as “long calls” Fig.1 and 2 and separate in Fig.3 and 4? More transparency in adding the omitted calls (even if not significant) comparisons as supplementary material would increase the value of the paper. Methods used in calculating call transitions need description and citations if done before.
- Figure 1 shows examples of USV in each category. I have problems in seeing the harmonic component of the “harmonic” type. The harmony in presented call is very weak on the spectrum and such falls under the possibility of being an artifact caused by the experimental setup. Also is harmonic a superimposing type? Almost every other type described in the paper could have technically a harmonic component.
Figure 3 is difficult to comprehend. Letters referring to sections of the figure do not match the description. Please add in description how to interpret the plots. On which axis are the follow-up calls? Should the probabilities in one line of the plot add up to 1?
Minor points:
38-39; There is a shift to continuous tense. If the authors state general phenomena the present simple tense should be used.
40; article “the” is redundant
43; past perfect tense would be more adequate
46 to 47; please add the subject to the sentence structure
48; “analysis of the mouse calls is not yet that far” - What does this mean? Far from what?
56-60; “component” - Component of what?
70; “a daily” is unnecessary
77; Add dimensions of the recording cage. Did the cage had also bedding in it?
129; ”significantly” – How significant was it?
131; “increased” – Was it significant?
136; “the red border” – Where is it? The border is grey in the manuscript.
138; “genotype p < 0.05; age p < 0.05”; different numbers in the figure
Figure 2; pie charts could use less vibrant colors
163, 168-9; “order probability “, “probability of one call followed the previous” -How is this calculated? Had it been done before?
166 (and other places); “mouse language” – Unwarranted usage of the word language. Better terms: “vocal repertoire”, “vocal communication” for example.
171; ”great differences” – How significant are they?
227, 236; “development of the communication” / changes in USV usage
223, 230; and many other places “Figure” “figure”; change all to “F”.
Figure 5; poor readability, dots would be better than spectrogram images
255; “precisely divided”, improper use, cited references have a range of 9-14 types
267; the length of the call not necessarily implies its complexity; a short call can be a highly modulated trill and a long call can be flat
Future insights:
For the future I would suggest the authors make video recordings of the “speed-dating” procedure and try to correlate calls with behavior. A partner preference test for female mice could also be beneficial to determine whether the KO mice are truly impaired vocally.
Author Response
First of all, we would like to thank the reviewers for the constructive and technically feasible points of criticism! We have addressed all queries raised by the reviewers and included the improvements in the revised manuscript. Please find here the detailed reply to your queries:
Reviewer #2:
Major points:
Query 1. The description of the subjects used requires revision. A more thorough characterization of the genetic background of the animals would be beneficial. From where were the original breeding pairs mice obtained? Where were the mutants generated? How many crossings with C57Bl/6 mice were there before the experiment? Please specify whether the WT and KO mice used in the experiment were siblings as it is unclear from description. Please place the information about the age of the animals during the recording sessions in M&M. Describe if the mice were naïve or experienced in male/female social encounters as this significantly alters their vocal repertoire. This information is presented later in the paper however M&M should contain such details. Furthermore, were the females checked for estrous cycle since the state that they were in could affect the outcome of the experiment?
Reply: The requested details were added to the extended M&M section (lines 66 – 112). Mice were bred on a mixed genetic background 129/Ola x C57Bl/6, in which the 129/Ola genetic background originally comes from the embryonic stem cells and the C57Bl/6 background from the blastocysts used for stem cell injection and the test crossing for germline transmission. To avoid possible inbreed effects, we generally keep our mouse strains in a mixed genetic background as an outbreed strain, independently of the genetic manipulation of the Spred2 alleles. Since humans are biologically outbreed individuals, breeding of mice on a mixed genetic background makes the obtained results more transferable to the human situation.
Query 2. Were the animals handled prior to the experiment? Were they familiarized with the “speed dating” setup? Did the analyzed recordings only include the 5 minutes when the male and female were together? In the described procedure the USV recordings should contain both male and female vocalizations at once, did the authors take that into consideration that the observed differences in groups may sex sensitive?
Reply: The requested details were added to the extended M&M section (lines 66 – 112).
Query 3. The authors have used DeepSqueak to analyze the USVs, specifically versions 2.5.2 and 2.6, these versions of the program do not allow free scrolling of the spectrogram. Only the newest 3.0 version incorporated this option from a GitHub fork branch. In a recent paper by Fonseca et al. (DOI:
10.7554/eLife.59161) compared several machine learning software in detection of mouse USVs. In their study DeepSqueak had an astounding 27.13% rate of missed calls, which from personal experience I find plausible. The recordings should be checked for missed calls. The call classification was done unsupervised by a second neural network. What was the input data used to train the network, were those the same 8000 images used for detection? I also suggest that after automatic classification the USVs types should be evaluated by a human since software does not guarantee perfect assignment of USV types.
Reply: We thank Reviewer 2 for this really interesting point! Indeed, initially we were not satisfied with the call recognition rate of the included DeepSqueak CNN. To improve this rate, we decided to train the network with calls of our mice, which might differ from mouse calls in other facilities or even from other continents!? The input data to train the CNN were the described 8000 random calls from wild-type mice, which were also re-evaluated by a human (Denis Hepbalsi). The calls included in this manuscript were recorded independently during the Speed Dating events. Following your suggestion to use DeepSqueak 3.0 to scroll through the spectrograms and look for missed calls revealed a rate of missed calls of 8.5%, which is in the range of other commonly used machine learning programs described in the Paper by Fonseca et al., which we included in the references (lines 114 – 121).
Query 4. In results multiple different variables are presented as proof of difference of KO mice. Please specify, how many parameter comparisons were made in total for each of the presented data? Why some calls in the presented results pulled together (flat, short, up) and others like step up are not, also complex and harmonic calls are once grouped as “long calls” Fig.1 and 2 and separate in Fig.3 and 4? More transparency in adding the omitted calls (even if not significant) comparisons as supplementary material would increase the value of the paper. Methods used in calculating call transitions need description and citations if done before.
Reply: Each call was examined further in terms of four characteristics: call length (in sec.), principal frequency (in kHz), delta frequency (in kHz) and mean power (in dB/Hz) (lines 129 – 131). This gives a total of forty parameter comparisons for the ten calls plus the comparisons of all calls taken together. Resulting significant differences with a clear focus on genotype-dependent changes are displayed e.g. in Figure 4. To avoid confusion of the reader, not significant parameters were not included in the manuscript. Furthermore, we have deleted the misleading plots of combined categories and – because of their similarity - we included the original category Harmonic into the category Complex.
Query 5. Figure 1 shows examples of USV in each category. I have problems in seeing the harmonic component of the “harmonic” type. The harmony in presented call is very weak on the spectrum and such falls under the possibility of being an artifact caused by the experimental setup. Also is harmonic a superimposing type? Almost every other type described in the paper could have technically a harmonic component.
Reply: Reviewer 2 is right! Only some calls might have a harmonic component but look otherwise like a Complex call. Since the harmonic components of such calls are sometimes hardly detectable, even manually, we decided to include this category in the Complex category and renamed the originally as Complex designated category to Frequency Steps. Unfortunately, there is no uniform nomenclature and definition existing, so far.
Query 6. Figure 3 is difficult to comprehend. Letters referring to sections of the figure do not match the description. Please add in description how to interpret the plots. On which axis are the follow-up calls? Should the probabilities in one line of the plot add up to 1?
Reply: We have corrected the corresponding text parts, added more information how to interpret the plots, and added the information that the follow-up calls are on the x-axis.
The probabilities should add up to 1 but do not reach this value completely since the values were rounded down and DeepSqueak eliminates calls with an appearance of less than one percent from the analyses.
Minor points:
Thanks again for carefully reading the entire manuscript and for the list of some mistakes in the first version of the manuscript! We have revised the entire manuscript carefully and tried correct all minor (and major) points of concern. Before resubmission, the revised manuscript underwent a spell-check by a native speaker.
Round 2
Reviewer 1 Report
The article has received major revisions which have sustainability improved the clarity of the manuscript. However, some concerns are noted by this reviewer
- USVs seemed to be recorded during the light portion of the light cycle, which is not typical of this kind of work. Please report as a limitation.
- The female estrous cycle was not properly controlled for as “attempted to use mice in the proestrus state” is not clear how many female mice were recorded in this stage and how many were not. Also, typically the estrus state should be used for mating paradigm recordings. The authors need to better clarify and justify these methods or list as a limitation of the study.
- Some grammar/word usage requires revisions. For example, ‘vocalise/vocalize’ and ‘to proof dumpness of the females.’
- The authors spend a lot of time discussing the advantages of training their own detection network. However, with the newest version of Deepsqueak, you can add undetected vocalizations, so their methods have no clear advantage over the standard version of Deepsqueak.
- The call classification needs to be cited. This is clearly based on Wright’s categories.
- The planned statistics still require to be more clearly stated in the methods before reported in the results such as the T-SNE plots.
- The discussion still needs to be tempered. The new combination of call types may not be an indication of matured vocal repertoire in old age. In fact, it could equally be due to a deterioration of the vocal system. A future study can investigate if these new call combination have positive behavioral outcomes and could support the hypothesis that the call repertoire is advancing with age, but this study currently cannot conclude this finding based on the data.
Reviewer 2 Report
See attached file.
